# Effects of Early Exposure of Isoflurane on Chronic Pain via the Mammalian Target of Rapamycin Signal Pathway

**DOI:** 10.3390/ijms20205102

**Published:** 2019-10-15

**Authors:** Qun Li, Reilley Paige Mathena, O’Rukevwe Nicole Eregha, C. David Mintz

**Affiliations:** Department of Anesthesiology and Critical Care Medicine, Johns Hopkins University School of Medicine, Baltimore, MD 21205, USA; qli21@jhmi.edu (Q.L.); rmathen1@jhmi.edu (R.P.M.);

**Keywords:** anesthesia neurotoxicity, neuropathic pain, mammalian target of rapamycin (mTOR), insular cortex (IC), anterior cingulate cortex (ACC), spinal dorsal horn (SDH)

## Abstract

Persistent post-surgical pain (PPSP) is a chronic pain condition, often with neuropathic features, that occurs in approximately 20% of children who undergo surgery. The biological basis of PPSP has not been elucidated. Anesthetic drugs can have lasting effects on the developing nervous system, although the clinical impact of this phenomenon is unknown. Here, we used a mouse model to test the hypothesis that early developmental exposure to isoflurane causes cellular and molecular alteration in the pain perception circuitry that causes a predisposition to chronic, neuropathic pain via a pathologic upregulation of the mammalian target of the rapamycin (mTOR) signaling pathway. Mice were exposed to isoflurane at postnatal day 7 and select cohorts were treated with rapamycin, an mTOR pathway inhibitor. Behavioral tests conducted 2 months later showed increased evidence of neuropathic pain, which did not occur in rapamycin-treated animals. Immunohistochemistry showed neuronal activity was chronically increased in the insular cortex, anterior cingulate cortex, and spinal dorsal horn, and activity was attenuated by rapamycin. Immunohistochemistry and western blotting (WB) showed a co-incident chronic, abnormal upregulation in mTOR activity. We conclude that early isoflurane exposure alters the development of pain circuits and has the potential to contribute to PPSP and/or other pain syndromes.

## 1. Introduction

Persistent post-surgical pain (PPSP) is defined as pain that continues for at least 3 months following a surgical intervention that either was not present prior to surgery or that is completely distinct in character from pre-operative pain at the surgical site [1,2]. PPSP is increasingly recognized as a common complication of pediatric surgery [3,4], with an estimated prevalence of 20% on the basis of a meta-analysis of 12 studies conducted by Rabbitts et al. [5]. The quality of pain in PPSP in adults varies by patient and is influenced by the type of surgery; it is often neuropathic in nature and there is evidence pointing to neural plasticity as a key underlying mechanism [6]. Although it is difficult to investigate the character of pain in children, two studies indicate that neuropathic pain symptoms are likely predominant in PPSP on the basis of questions posed to the child and parent [7,8], and, interestingly, a study of PSPP after hernia repair showed hyperalgesia to pin-prick [9]. The risk factors for PPSP that have been identified by meta-analysis include parental factors as well as pre-operative pain, anxiety, and coping ability [5], but there is currently no insight into the biological basis for PPSP.

Nearly all children who undergo surgery receive general anesthesia, and there is increasing reason to believe that the commonly used anesthetic drugs may have unintended effects on the development of the nervous system [10]. To date, most investigation of anesthetic effects has focused on potential adverse effects on cognitive function because this was the first effect demonstrated in rodent models [11], but there is increasing interest in the idea that the most clinically significant effects of anesthetic neurotoxicity may be manifested in other functional domains [12]. For example, two non-human primate studies have demonstrated increased reactivity to stressful stimuli [13,14], and both human and non-human primate studies have found evidence of deficits in motor function [15,16]. The mechanisms by which anesthetics act on the developing nervous system to effect lasting change have not been elucidated, but alterations in synapse number, distribution, and function have been identified as likely contributors [17,18,19,20] Given the role of synaptic plasticity in the onset of chronic pain [21], there exists the intriguing possibility that exposure to certain anesthetics might constitute a biological risk factor for PPSP.

In this study, we employed a mouse model to test the overall hypothesis that early developmental exposure to isoflurane, one of the most commonly used general anesthetic agents, can cause an increase in the risk of chronic pain, with a goal of understanding whether anesthetic technique may be an important factor in PPSP. We focused on two cortical regions that are involved in pain transduction and modulation—the insular cortex (IC) and anterior cingulate cortex (ACC)—and on the spinal cord dorsal horn (SDH). IC and ACC have been implicated as important centers for pain information memory [22,23,24,25] and SDH is a critical area for regulation and transmission of pain signal [26]. In our previous work, we have found that early developmental isoflurane exposure aberrantly increases activity in the mammalian target of rapamycin (mTOR) pathway [27,28,29], which is a complex system that interpolates between extracellular cues and intracellular signaling systems that regulates a variety of biological processes [30,31], including central nervous system (CNS) development [32]. Activity in the mTOR pathway has been implicated in chronic pain generally and in neuropathic pain specifically [33,34,35,36], and thus we further explored the possibility that early developmental isoflurane exposure may lead to a lasting effect on pain pathways via an action on the mTOR pathway.

## 2. Results

### 2.1. Effect of Early Isoflurane Exposure on Chronic Pain Behaviors

For all experiments in this study, three groups were compared as follows: naïve control, isoflurane exposure plus vehicle, and isoflurane exposure plus rapamycin treatment (*n* = 12 per group). We first asked whether early isoflurane exposure and rapamycin treatment had a lasting effect on chronic pain behaviors: (1) The tail flick test evaluated pain response to heat stimulation. Our data showed that isoflurane exposure at P7 caused a significant decrease in tail flick latency (10.11 ± 1.82 vs. 7.56 ± 1.39 s; *p* < 0.01) to the light beam at P63. This alteration of response to nociceptive stimuli was ameliorated with rapamycin treatment (10.07 ± 1.8 s; *p* < 0.01) (Figure 1B). (2) Mechanical allodynia and hyperalgesia were tested with von Frey filament application. Hind-paw withdrawal threshold in isoflurane exposed mice was significantly lower compared to naïve control (1.88 ± 0.15 vs. 1.65 ± 0.21 g; *p* < 0.05), and thresholds were restored to levels not significantly different from control with rapamycin injection (1.85 ± 0.19 g; *p* < 0.05) (Figure 1C). (3) After formalin microinjection, animals in each of the three study groups spent almost identical time licking legs or paws in phase I (0–5 min). However, in phase II (15–30 min), isoflurane-exposed mice had a longer cumulative leg/paw-licking time than control (232.5 ± 65.15 vs. 311.5 ± 54.64 s; *p* < 0.01), and rapamycin appeared to reverse this effect (248.5 ± 57.19 s; *p* < 0.05) (Figure 1D).

### 2.2. Effect of Isoflurane Exposure on Expression of mTOR Pathway and Neuronal Activity in Insular Cortex (IC)

In order to evaluate the effect of isoflurane on neuronal activation in IC, we first conducted quantitative fluorescence immunohistochemistry (IHC) in brain sections using an antibody against c-fos, which is expressed in neurons following depolarization and represents a marker of neuronal activity. The location of IC was identified according to criteria from Paxinos and Franklin’s Mouse Brain Atlas [37] (red box in Figure 2A). Early isoflurane exposure resulted in a greater than three-fold increase of c-fos-labeled neurons (21.7 ± 14.78 vs. 78.64 ± 18.31/mm^2^; *p* < 0.001) in IC, and rapamycin injection reversed this effect (39.42 ± 13.12/mm^2^; *p* < 0.01) at postnatal week 9 (Figure 2B). Next, IHC was performed to detect the expression of phospho-s6 (pS6), a reporter of mTOR activity. We found the number of pS6 positive cells in IC (mainly in layer 3) of isoflurane-exposed mice was greater than in the control (73.76 ± 23.89 vs. 165.13 ± 45.55/mm^2^; *p* < 0.01), and rapamycin treatment decreased this number (92.04 ± 23.18/mm^2^; *p* < 0.01) (Figure 2C). Considering the data from pS6 and NeuN double-labeling in which almost all of the pS6 positive cells were neurons in IC (high power image in Figure 2C), quantitative analysis was performed for pS6 labeling only. To further confirm these results, we conducted western blotting (WB) using extracts from the IC. The level of phosphorylated mTOR (p-mTOR) was examined. The ratio of p-mTOR band intensity over total mTOR (t-mTOR) was dramatically elevated with isoflurane exposure compared to the control (29.79 ± 9.56% vs. 104.66 ± 21.42%; *p* < 0.0001) and a significant recovery resulted from rapamycin treatment (58.63 ± 21.7%; *p* < 0.01) (Figure 2D). However, the ratio of t-mTOR intensity over β-actin was identical for all groups (110.5 ± 12.73% vs. 113.17 ± 13.01% vs. 108.07 ± 20.99%; no significant difference) (Figure 2E). Given the role of mTOR in synaptic development and plasticity in the CNS, we hypothesized that isoflurane exposure alters expression of synaptic proteins known to be involved in chronic pain. To test this, we measured the level of post-synaptic density 95 (PSD95) in IC by WB. Isoflurane exposure enhanced the reactive intensity of PSD95 (57.77 ± 16.39% vs. 107.36 ± 19.1%; *p* < 0.01) and rapamycin ablated this effect (75.85 ± 18.44%; *p* < 0.05) (Figure 2F).

### 2.3. Effect of Early Isoflurane Exposure on Expression of mTOR and Neuronal Activity in Anterior Cingulate Cortex (ACC)

The ACC, a cortical region in the prefrontal cortex (PFC), is located above the corpus callosum and beside the central sulcus. It was identified using the same brain atlas as mentioned above (Figure 3A). To test the alteration of neuronal activation and mTOR expression in ACC after anesthesia exposure and rapamycin treatment, we examined c-fos-positive and pS6-immunolabeled cells in this region. We counted 70.33 ± 19.83/mm^2^ c-fos-positive neurons in the control and 102 ± 14.48/mm^2^ in isoflurane-exposed animals (*p* < 0.01). This number was reversed to 74.5 ± 11.4/mm^2^ in the isoflurane plus rapamycin injection group (*p* < 0.05) (Figure 3B). For the activity level of mTOR signal pathway in ACC, we found a significant increase of pS6 positive cells in isoflurane exposure (68.5 ± 21.92/mm^2^) compared to control conditions (40 ± 10/mm^2^; *p* < 0.05), which was restored with rapamycin treatment (44.17 ± 8.33/mm^2^; *p* < 0.05) (Figure 3C). The pS6 positive cells in ACC, the same as those in IC, were confirmed as neuronal cells with pS6/NeuN double immunostaining (data not shown). Isoflurane exposure increased the expression of p-mTOR (over t-mTOR) compared to the control (115.17 ± 12.43% vs. 68.67 ± 14.29%; *p* < 0.001) and rapamycin attenuated the p-mTOR/t-mTOR ratio (85.5 ± 13.53%; *p* < 0.01) (Figure 3D). Isoflurane exposure and administration of rapamycin did not significantly affect the level of total mTOR in ACC (86.88 ± 12.01% vs. 84.35 ± 15.39% vs. 95.85 ± 21.6%; no significant difference) (Figure 3E). The reactive density of PSD95 was enhanced with isoflurane exposure (66.33 ± 13.91% vs. 97.5 ± 16.67%; *p* < 0.01) and the alteration was reversed by rapamycin treatment (73.5 ± 11.73%; *p* < 0.05) (Figure 3F).

### 2.4. Effect of Isoflurane Exposure on Neuronal Activity and pS6 Expression in Superficial Spinal Dorsal Horn (SDH)

The alteration of neuronal activity in dorsal spinal cord after isoflurane exposure and rapamycin treatment was examined. Both c-fos-positive and pS6-positive cells were distributed throughout the whole spinal cord, but we only performed quantitative analysis in superficial SDH (lamina I and II) because it is the critical area for transmission and modulation of pain signaling. Compared to control, isoflurane exposure upregulated the number of c-fos positive neurons (29.65 ± 9 vs. 54.75 ± 16.2/mm^2^; *p* < 0.05) and rapamycin treatment restored this number near the control (32.3 ± 9.29/mm^2^; *p* < 0.05) (Figure 4A). pS6-immunolabeled cells were also detected in superficial SDH. Considering a certain number of pS6-positive cells looked like glial cells (smaller size and multipolar shaped), we only measured pS6-positive/NeuN-positive double-labeling to evaluate the neuronal expression of mTOR in this region. The number of pS6-positive/NeuN-positive neurons in isoflurane-exposed mice (126.67 ± 26.88/mm^2^) was greater than the control (89.67 ± 14.53/mm^2^; *p* < 0.05), and this number was attenuated by rapamycin injection (90.17 ± 14.11/mm^2^; *p* < 0.05) (Figure 4B).

## 3. Discussion

In the present study, we first report that early isoflurane exposure increased the likelihood of chronic neuropathic pain, including effects on mechanical allodynia, thermal allodynia, and inflammatory pain (Figure 1B–D). Isoflurane simultaneously induced increased mTOR pathway activity and increased neuronal activity in the IC, ACC, and SDH, all of which are regions known to be involved in pain processing. Inhibition of mTOR with rapamycin treatment substantially ameliorated the functional reactions to pain stimuli that were associated with early isoflurane exposure. These findings suggest that exposure to isoflurane and other similar anesthetic agents have the potential to be risk factors for the development of PPSP.

To explain our behavioral findings, we have tested for biological evidence of changes in CNS regions that are known to be involved in pain processing. The IC and ACC not only play an important role in complex cognitive functions, such as learning, memory, and awareness of feelings [37,38,39], but also contribute to neuropathic pain perception [40,41]. Synaptic plasticity in IC and ACC is critically important to the induction of chronic pain [42]. The superficial SDH (lamina I and II) is a critical region for transmission of pain signal from the primary afferent pathway and regulation of nociceptive sensation [26]. In addition, a neural top-down connection between IC or ACC and SDH, as a “descending facilitation system”, modulates pain signaling. Selective activation of ACC neurons sensitizes projecting neurons located in SDH and causes behavioral pain sensitization [25]. We assayed neuronal activity in these areas using c-fos, the product of proto-oncogene c-fos, which is rapidly expressed in the nociceptive information transmission pathway under neuropathic conditions [26,43]. Although our findings of increased neuronal activity in these regions do not prove a causal relationship between the observed increase in chronic pain measures, they are highly suggestive.

The mTOR protein is an obligatory component of two complexes (mTORC1 and mTORC2) which are at the center of a signaling system that regulates cellular activities including proliferation, differentiation, apoptosis, metabolism, transmitter release, synaptic formation, and other biological processes [31]. In the intact rat model, mTOR and several downstream molecules, such as eukaryotic inhibition factor 4E-binding protein (4E-BP) and p70 ribosomal S6 protein kinase (p70S6K), were expressed in IC, ACC, SDH, and dorsal root ganglion (DRG); however, the level of *p*-mTOR was a very low level [22,24,44]. It has been reported that in a variety of pathological conditions, such as injury or nociceptive stimulation that stimulate pain, that the activity of mTOR pathway molecules, as evidenced by phosphorylation of mTOR, is dramatically increased [22,24,45], a finding that our results have replicated. Early isoflurane exposure increased the number of immunolabeled neurons for pS6, a reliable downstream reporter for mTOR pathway activity, in IC, ACC, and superficial DSH (Figure 2, Figure 3 and Figure 4). Considering the correlation between enhancement of mTOR pathway molecules and generation of pain, our findings suggest a possible mechanism by which early exposure to isoflurane might contribute to chronic pain. Using WB, expression of p-mTOR dramatically increased following isoflurane, but t-mTOR was not significantly altered among different groups (Figure 2D and Figure 3D). This indicates that the predominant internal cellular reaction is the increased phosphorylation of mTOR and activation of mTOR pathway molecules, rather than increased expression of the mTOR protein. We also observed an increased number of c-fos-immunolabeled neurons in IC (Figure 2B), ACC (Figure 3B), and superficial DSC (Figure 4A) in isoflurane-exposed cases. Restoration of c-fos-positive neuron number by rapamycin treatment to normal levels provided further support for the argument that the mTOR pathway may be involved in chronic pain activation, although we cannot exclude the possibility that rapamycin could be acting on other targets or on the mTOR pathway in other CNS regions. Our data do not fully distinguish between effects mediated by mTORC1 and MTORC2, as the rapamycin paradigm used here was expected to inhibit both pathways [46], and previous work on this topic suggests that isoflurane acts simultaneously to upregulate both of these pathways [28]. Therefore, our results should be interpreted as an effect mediated by both complexes until further experimentation can provide greater insight.

PSD95 is an essential scaffold protein in the post-synaptic component of excitatory synapses [47]. It has been shown that PSD95 contributes the maintenance of neuropathic pain, and knockdown of PSD95 in spinal cord delays the development of chronic neuropathic pain [48]. Under stress conditions, PSD95 was upregulated in the hippocampus in an mTOR-dependent pattern [49], and increased PSD95 was detected in IC and ACC following the peripheral never injury [22,24]. In the present study, the level of PSD95 expression was elevated in IC and ACC by early isoflurane exposure and reversed with rapamycin (Figure 2E and Figure 3E). These data suggest a very interesting future direction to investigate potential mechanisms by which isoflurane and other anesthetics could increase the risk of PPSP in children who undergo surgery.

In conclusion, early isoflurane exposure produces pathological alteration in CNS circuits and aberrantly increases the activity of the mTOR signaling pathway in areas related to nociceptive sensation. These studies have all the usual caveats associated with the limitations of rodent models of pain and of anesthesia toxicity. We believe the significance of our findings is that it should prompt research in two further directions: (1) It would be very instructive to examine retrospective datasets of patients who developed PPSP and matched patients who did not in order to test for differences in anesthetic technique that might be suggestive of a potentially modifiable risk factor for PPSP related to anesthetic choices. (2) We believe that further mechanistic investigation of anesthetic effects on the development of chronic pain are independently interesting in uncovering potential mechanisms to prevent PPSP, contributing to our understanding of the development of pain circuitry and its relationship with chronic pain, and also to better describe the putative effects of anesthetics on the developing nervous system.

## 4. Materials and Methods

### 4.1. Animal Paradigm and Experimental Timeline

C57BL/6 mice were used in this study. Sex was not factored into the research design as a biological variable and both sexes were equally represented. All study protocols involving mice were approved by the Animal Care and Use Committee at the Johns Hopkins University, and were conducted in accordance with the NIH guidelines for care and use of animals (approved No. MO17M229; approved date: 24 August 2017). At P7, animals were exposed to isoflurane or room air for 4 h. From P21 to P35, isoflurane-exposed mice were injected (i.p.) with rapamycin or with vehicle, on the basis of previous work in our lab showing that this is an effective time to intervene in the mTOR pathway [50]. From P56–P62, all mice underwent behavior tests for chronic pain. Animals were sacrificed at P63, and brains and lumbar spinal cords were harvested for immunohistochemistry and WB (Figure 1A).

### 4.2. Isoflurane Exposure

Using previously described methods, 36 mice were investigated in total [50]. At P7, 24 littermates were randomly selected for isoflurane exposure and 12 mice remained in room air as naïve controls. Volatile anesthesia exposure was accomplished using a Supera tabletop portable non-rebreathing anesthesia machine (Supera. Clackamas, OR, USA). A total of 3% isoflurane mixed in 100% oxygen as carrier gas (CG) was initially delivered in a closed chamber for 3–5 min, and after loss of righting reflex, animals were transferred to the specially designed plastic tubes. A heating pad (36.5 °C) was placed underneath the exposure tubes. The mice were exposed to 1.5% isoflurane carried with CG for 4 h. A calibrated flowmeter was used to deliver CG at a flow rate of 5 L/min, and an agent-specific vaporizer was used to deliver isoflurane. After isoflurane exposure, mice were returned to their dams together with their littermates upon regaining righting reflex [51].

### 4.3. Rapamycin Injection

The 36 mice were equally divided into 3 groups (*n* = 12 per group): (1) control; (2) isoflurane exposure plus vehicle; and (3) isoflurane plus rapamycin injection. From P21 to P35, 12 isoflurane-exposed mice (group 3) were injected (i.p.) twice daily with 100 µL 0.2% rapamycin dissolved in vehicle solution, and 12 were injected with vehicle only (group 2). Vehicle consisted of 5% Tween 80 (Sigma-Aldrich, St. Louis, MO, USA), 10% polyethylene glycol 400 (Sigma-Aldrich, St. Louis, MO, USA), and 8% ethanol in saline. No treatment was used for control animals (group 1) [51].

### 4.4. Behavior Tests

Pain behavior tests were performed at postnatal week 9 for all groups. There was at least a 48 h time interval between different tests.

#### 4.4.1. Tail Flick Test

First, the animals were restrained in a plexiglass tube and placed on the tail flick apparatus (IITC Inc., Woodland Hills, CA, USA). A noxious beam of light was focused on the tail, 2 cm from the tip. When light stimulation began, the timer started. The timer stopped automatically while the animal flicked its tail, and the recorded time (latency) was a measure of the pain threshold. The intensity of the radiant heat source for the control mice was adjusted to yield baseline latencies between 9 and 13 s. In order to minimize injury in the animals, a cut-off time of 20 s was used. Each animal was tested 5 times with at least a 30 min time interval. The average value (second) was recorded as final time latency for tail flick.

#### 4.4.2. The von Frey Test

Mechanical hyperalgesia in the mice was determined by applying von Frey filaments (Smith and Nephew Inc., Germantown, WI, USA) to the plantar surface of the hind-paws. Before being tested, animals were acclimatized for 60 min to the testing environment, which featured individual plexiglas cubicles over a coated wire mesh platform. A series of calibrated von Frey filaments (0.04–2.0 g) were applied for 5 s with enough force to cause filament buckling, and withdrawal response of the hind-paw was observed. The smallest filament to evoke ≥3 withdrawal responses out of five repeated applications was determined, and the average value (gram) from both hind paws was recorded as the withdrawal threshold.

#### 4.4.3. Formalin Test

Prior to testing, animals were adapted to the testing environment for at least 1 hour. The mice were placed individually in a transparent plastic cage (30 × 30 × 36 cm), which served as the observation chamber. The mice were able to walk freely. In order to see the paw movement easily, a mirror was obliquely placed underneath the chamber. After this adaption period, 10 µL 2% formalin (Fisher Scientific, Hampton, NH, USA) (in saline) was injected interplantarly into the left hind paw using a microsyringe with a 26-gauge needle. The observation period started immediately after formalin injection. The total amount of time the animal spent licking the injected paw or leg was recorded in phase I (neurogenic phase, 0–5 min) and phase II (inflammatory phase, 15–30 min) after formalin injection.

### 4.5. Immunohistochemistry (IHC)

At P63, six mice from each group (*n* = 6) were perfused with 40 mL 4% paraformaldehyde in phosphate-buffered saline (PBS). Brains and L4–6 lumbar spinal cords were removed, post-fixed overnight, and cryo-protected in 25% sucrose for 48 h. The brain tissue was coronally sectioned in 40 µm using a freezing microtome, and the spinal cord was cut in 20 µm sections with a cryostat. Randomly selected brain sections from each group were processed for floating IHC, and spinal cord sections were immunostained on glass slides. First, sections were blocked in 10% normal goat serum (Sigma Aldrich, St. Louis, MO, USA) and 0.1% triton X-100 for 60 min, followed by primary antibody incubation at 4 °C overnight. In this study, rabbit anti-c-Fos (1:1,000; Cell Signaling, Danvers, MA, USA) and rabbit anti-phospho-S6 (pS6) (1:1,000; Cell Signaling, Danvers, MA, USA) primary antibodies were used. For some sections, rabbit anti-pS6 mixed with mouse anti-NeuN (1:500; Millipore, Burlington, MA, USA) was applied for double-labeling. After 3 × 10 min washes in PBS, sections were incubated with secondary antibodies—Cy3 (1:600; Jackson Laboratory, Bar Harbor, ME, USA) or Alexa 488 conjugated (1:600; Jackson Laboratory, Bar Harbor, ME, USA) goat anti-rabbit IgG and Alexa 488-goat anti-mouse IgG (1:300; Jackson Laboratory, Bar Harbor, ME, USA) for 2 h. After 3 × 10 min PBS washes, brain sections were mounted onto slides. Both brain and spinal cord sections were coverslipped after air drying. Then, they were imaged using a Leica 4000 confocal microscope (Leica, Wetzlar, Germany). All c-fos-positive (labeled in nuclei) and pS6-immunolabeled neurons in IC, ACC, and superficial SDH (lamina I and II) were counted and quantitatively analyzed using the ImageJ program (NIH, v.1.52p, Bethesda, MD, USA). Double-labeled cells were identified in channel-merged images [51].

### 4.6. Western Blotting (WB)

A total of six animals per group (*n* = 6) were quickly perfused with cold saline on day 63. Brains were immediately removed and flash frozen in dry ice. IC and ACC were isolated with punch biopsies according to Paxinos and Franklin’s Mouse Brain Atlas [52]. Tissue from IC and ACC was lysed in the lysis buffer and homogenized with a bullet bender (Next Advance, Troy, NY, USA). Samples were prepared with 1:1 denaturing sample buffer (Bio-Rad, Hercules, CA, USA), boiled for 5 min, and run on Novex 4–12% Bis-Tris Protein Gels (Thermo Fisher, Waltham, MA, USA) in NuPage running buffer (Thermo Fisher, Waltham, MA, USA) with 150 volts for about 1 h. The proteins were transferred to Novex nitrocellulose-blotting membranes (Thermo Fisher, Waltham, MA, USA). Blots were probed with the following antibodies: anti-*p*-mTOR (1:1000, Cell Signaling, Danvers, MA, USA), anti-t-mTOR (1:1000, Cell Signaling, Danvers, MA, USA), anti-PSD95 (1:1000, Cell Signaling, Danvers, MA, USA), and anti-β-actin (1:2500, Cell Signaling, Danvers, MA, USA). Blots were visualized using Pierce Biotechnology ECL western blotting substrate kit (Thermo Fisher, Waltham, MA, USA). Images were acquired using ChemiDoc imaging system (BioRad, Hercules, CA, USA) and were quantitated with ImageJ (NIH, v.1.52p, Bethesda, MD, USA). The ratios of band density from p-mTOR over t-mTOR or PSD95 over β-actin were calculated, respectively, and quantitative analysis was performed with the ImageJ program [51]. Each band that is shown in the figures represents a single trial with a unique animal.

### 4.7. Statistical Analysis

Statistical comparisons were performed with the Prism (GraphPad, v.8, La Jolla, CA, USA) program. Data were analyzed using a two-tailed one-way analysis of variance (ANOVA), except the formalin test, which used a two-way ANOVA. A post-hoc Tukey’s test was used for inter-group comparisons (control vs. isoflurane plus vehicle vs. isoflurane plus rapamycin). The criteria for significant difference was set a priori at *p* < 0.05. Results were expressed as mean ± standard deviation (SD) [51].

## Figures and Tables

**Figure 1 ijms-20-05102-f001:**
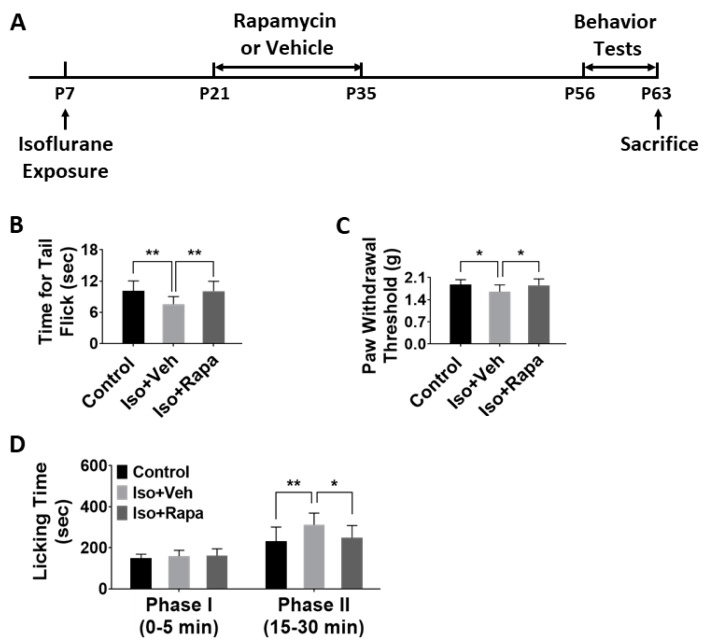
(**A**) Experimental timeline. At P7, two-thirds of total mice were exposed to isoflurane for 4 h and one-third of animals remained in room air as naïve control. From P21 to P35, isoflurane-exposed mice were treated with rapamycin or vehicle at 48 h intervals. The pain behavior tests were performed at P56-P62. All animals were sacrificed at P63 for immunohistochemistry (IHC) and western blotting (WB). (**B**) Tail flick test. Early isoflurane exposure caused significant decrease of tail flick latency as light beam was applied on tail. This decrease was antagonized with rapamycin treatment (one-way ANOVA). (**C**) von Frey filament test. Hind paw withdrawal threshold in isoflurane-exposed mice was significantly lower than control, which was restored with rapamycin injection (one-way ANOVA). (**D**) Formalin test. After formalin microinjection, animals in three groups spent almost identical cumulative time to lick legs or paws in phase I (0–5 min). In phase II (15–30 min), isoflurane-exposed mice had a longer licking time than control, and rapamycin reversed this effect (two-way ANOVA). Iso: isoflurane; Veh: vehicle; Rapa: rapamycin. *: *p* < 0.05; **: *p* < 0.01. Error bars: SD.

**Figure 2 ijms-20-05102-f002:**
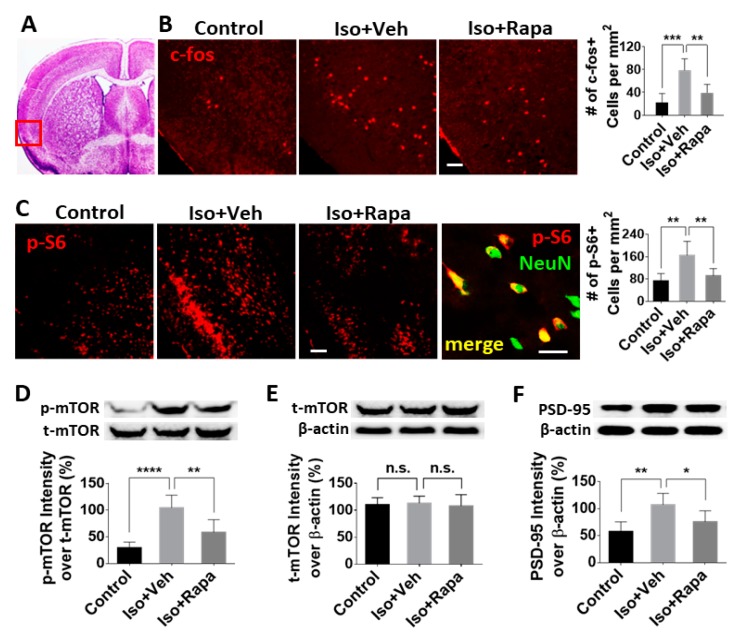
Effect of isoflurane exposure on neuronal activity in the insular cortex (IC). (**A**) Representative coronal brain section. The red box indicates the location in Figures B and C. (**B**) Immunohistochemical (IHC) image. Early isoflurane exposure resulted in dramatic increase in number of c-fos-labeled neurons in IC, and rapamycin injection reversed this effect. Bar = 50 μm. The histogram showed quantitative results. (**C**) The number of phospho-s6 (pS6) positive cells in isoflurane-exposed mice was revealed to be greater than in the control. Notice the dramatic increase of pS6 positive cells in layer 3. Rapamycin injection decreased this number. Bar = 100 μm. The high-power double IHC image for pS6 and NeuN indicated that these pS6 positive cells in IC were neurons. Bar = 50 μm. Graph showed quantitative results. (**D**) The mammalian target of rapamycin (mTOR) phosphorylation was examined with WB using IC tissue. The ratio of band intensity of phosphorylated mTOR (p-mTOR) over total mTOR (t-mTOR) was dramatically elevated by isoflurane exposure and a recovery resulted from rapamycin treatment. (**E**) Isoflurane exposure and rapamycin injection did not alter the level of total mTOR in IC. (**F**) Expression of post-synaptic density 95 (PSD95) in IC. WB shows the ratio of PSD95-positive bands over β-actin was significantly upregulated by isoflurane and returned to near control levels with rapamycin treatment. Statistics for all tests in this figure were one-way ANOVA. Iso: isoflurane; Veh: vehicle; Rapa: rapamycin. n.s.: no significance; *: *p* < 0.05; **: *p* < 0.01; ***: *p* < 0.001. Error bars: SD.

**Figure 3 ijms-20-05102-f003:**
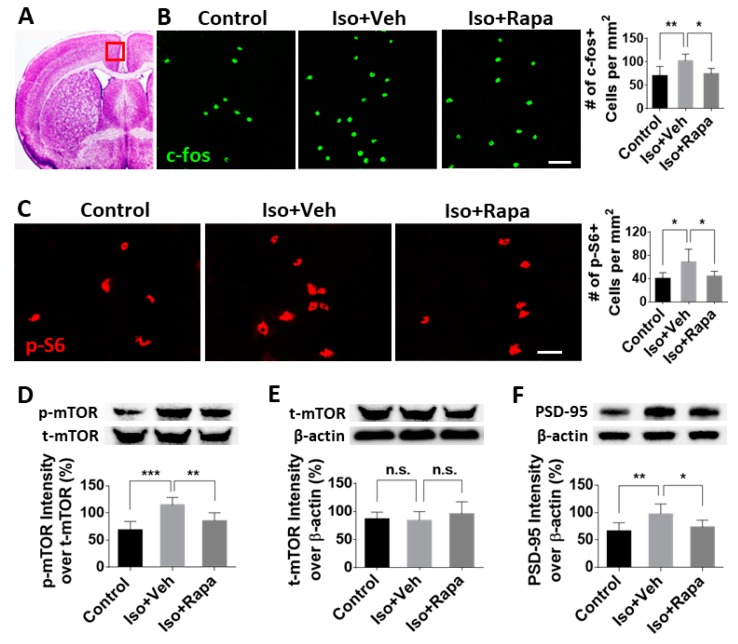
Neuronal activity and mTOR level in the anterior cingulate cortex (ACC). (**A**) Coronal brain section with ACC. Red box represents the location in Figures B and C. (**B**) Isoflurane upregulated the number of c-fos-positive neurons in ACC and rapamycin reversed this effect. Bar = 50 μm. (**C**) Number of pS6-immunolabeled cells in isoflurane-exposed mice was greater than in the control. Rapamycin decreased this number. Bar = 50 μm. (**D**) WB data showing phosphorylation of mTOR in ACC. The ratio of p-mTOR intensity over t-mTOR was enhanced by isoflurane exposure and reversed with rapamycin treatment. (**E**) The ratio of t-mTOR over β-actin was identical in all groups. (**F**) PSD95 level in ACC. WB showed the ratio of PSD95-positive band over β-actin was significantly upregulated by isoflurane and decreased close to the control level with rapamycin treatment. One-way ANOVA for all cases. Iso: isoflurane; Veh: vehicle; Rapa: rapamycin. n.s.: no significance; *: *p* < 0.05; **: *p* < 0.01; ***: *p* < 0.001. Error bars: SD.

**Figure 4 ijms-20-05102-f004:**
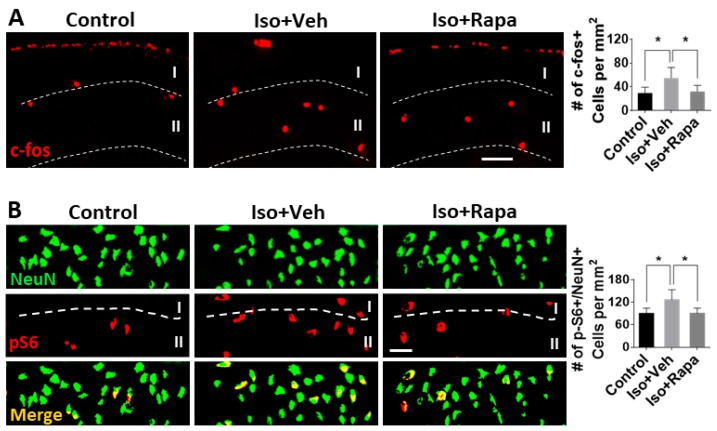
Neuronal activity and pS6 expression in superficial spinal dorsal horn (SDH; lamina I + II). (**A**) Isoflurane exposure upregulated the c-fos-positive neurons and rapamycin treatment restored this number near the control. Bar = 50 µm. Graph showed quantitative results. (**B**) The number of pS6-positive/NeuN-positive neurons in lamina I + II of isoflurane-exposed mice was greater than the control, and was attenuated with rapamycin treatment. Bar = 50 µm. One-way ANOVA for both (**A**) and (**B**). Iso: isoflurane; Veh: vehicle; Rapa: rapamycin. *: *p* < 0.05. Error bars: SD.

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
