# Peer review of "Effects of Early Exposure of Isoflurane on Chronic Pain via the Mammalian Target of Rapamycin Signal Pathway"

_ijms, 2019, doi:10.3390/ijms20205102_

Round 1
Reviewer 1 Report
COMMENTS TO AUTHORS:
Title: Effects of early exposure of isoflurane on chronic pain via the mammalian target of rapamycin signal pathway.
This research article try to demonstrate the association between persistent post-surgical pain (PPSP) with early exposure to isoflurane and mTOR pathway. Results from behavioral test realized two month after 4 h to isoflurane exposure, point out changes in the sensibility to pain stimulus in mice under early isoflurane exposure, that did not happen in rapamycin-treated mice. Using immunofluorescence and western blot approximations, the authors show an increase in neuronal activity (measured using c-fos+ cells) and an increment in the levels of p-S6 (a substrate of pS6K a downstream element of mTORC1) in isoflurane mice group respect to control. In parallel shows an increase in levels of p-mTOR and PSD-95. They conclude that early isoflurane exposure alters pain circuits and contribute to development PPSP by alteration of mTOR pathway.
The manuscript is written correctly and the results are consistent as a whole but in my opinion they are some important aspect that will improve this paper.
1.- The dose and timing of rapamycin used in this work is enough to inhibit both complex of mTORC: mTORC1 and mTORC2 (Mukhopadhyay S et al. Mol Cancer Ther March 1 2016 (15) (3) 347-353; DOI: 10.1158/1535-7163.MCT-15-0720). The authors should discuss their results based on this fact, not only based on the mTOR protein, which is always being part of one of these two complexes.
2.- In figures 2-4 the results shows a number of c-fos+ neurons after early isoflurane exposure or even in the control, that not correspond with the number of pS6+ cells, in fact in IC and SDH the number of pS6+ cells is double of c-fos+ cells and in ACC the results shows the opposite distribution of both markers. The authors should explain this apparent discrepancies.
It would will be interesting a double immunofluorescence using cfos and pS6 in order to clearly demonstrate, the relationship betwen neuronal activity and activation of mTORC1 pathway.
3.- Why the authors have not injected rapamycin in mice unexposed to isoflurane?
4.- In relation with the WB results, is not clear in the text the origin of the bands. Each band: Is a pool of lysates from different mice of the same experimental group? Or each band belong to one representative animal?. The authors should show a WB with the bands corresponding to all the animals used in the experiment separately, in order to observe the variations between mice (intragroup).
Is necessary a load control with β-actin or β-tubulin in the wb of t-mTOR and p-mTOR, to demonstrate that t-mTOR does not vary in each samples.
Author Response
Point 1.- The dose and timing of rapamycin used in this work is enough to inhibit both complex of mTORC: mTORC1 and mTORC2 (Mukhopadhyay S et al. Mol Cancer Ther March 1 2016 (15) (3) 347-353; DOI: 10.1158/1535-7163.MCT-15-0720). The authors should discuss their results based on this fact, not only based on the mTOR protein, which is always being part of one of these two complexes.
The reviewer raises an excellent point, which we have been attempted to address in other experimental work but neglected to clarify in this manuscript. Using mTORC1 and 2 specific pathway inhibitors and activity markers we have previously shown (somewhat to our surprise) that isoflurane appears to act on both pathways (Xu IJMS 2018 doi: 10.3390/ijms19082183). We have amended the discussion to make it clear that the effects we have demonstrated should be viewed as being mediated by both complexes and have included the reference above as well as the one suggested by the reviewer.
Point 2.- In figures 2-4 the results shows a number of c-fos+ neurons after early isoflurane exposure or even in the control, that not correspond with the number of pS6+ cells, in fact in IC and SDH the number of pS6+ cells is double of c-fos+ cells and in ACC the results shows the opposite distribution of both markers. The authors should explain this apparent discrepancies.
While c-fos labeling has been shown to be a reliable reporter of neuronal activity, it does have some notable limitations, chief of which is that it can only be interpreted in relative rather than absolute terms. Two factors are likely at work: 1. Care was taken that the timing of sacrifice relative to experimental intervention was as close to identical as possible within each experiment that comprises as figure. However, between figures this timing varied, and this likely accounts for some or all of the baseline difference in c-fos labeling that the reviewer noted. 2. The labeling for c-fos has tendency to fade. All groups within the same figure had a very similar time between immunohistochemistry and imaging, but the time varied between figures, which may also have affected the baseline. We have not altered the text of the manuscript related to this comment, but are happy to do so if it is felt to be important.
It would will be interesting a double immunofluorescence using cfos and pS6 in order to clearly demonstrate, the relationship betwen neuronal activity and activation of mTORC1 pathway.
We absolutely agree with the reviewers and are currently doing follow-up experiments to clarify this point. Our work at present does not make it clear whether the effects on the mTOR pathway are actually in the neurons with altered activity patterns, or whether they are actually in surrounding cells in the network that constitute important inputs or perhaps even in glial cells that modulate the affected neurons.
Point 3.- Why the authors have not injected rapamycin in mice unexposed to isoflurane?
In previous work, shown in S5 of Kang et al. PLoS Bio 2017 (doi.org/10.1371/journal.pbio.2001246) we have tested for an effect of rapamycin alone and found none. That manuscript related to a learning and memory effect occurring in the hippocampal dentate gyrus, and thus it is possible that it would not translate to this context. In future expanded work on this topic we plan to include a “rapamycin only” control to allow for optimal interpretation of our findings related to pain pathways.
Point 4.- In relation with the WB results, is not clear in the text the origin of the bands. Each band: Is a pool of lysates from different mice of the same experimental group? Or each band belong to one representative animal? The authors should show a WB with the bands corresponding to all the animals used in the experiment separately, in order to observe the variations between mice (intragroup).
We have amended the methods text to clarify that each band which is shown represents an individual animal, which is representative of the whole group that is quantified. Given that we have quantified these data and the variations are presented in the error bars, we would prefer not to show all the bands for the entire experiment in the figure, as we think it will be confusing to the reader, however we are happy to do so if the editor feels it is imperative.
Is necessary a load control with β-actin or β-tubulin in the wb of t-mTOR and p-mTOR, to demonstrate that t-mTOR does not vary in each samples.
Some similar published work using this approach showed only t-mTOR as the control, since it is the relative change between p-mTOR and t-mTOR is what is being tested, and so we had not originally included the loading control. However, we have included this in the revised manuscript to address the reviewer’s concern.
Reviewer 2 Report
The study entitled, “Effects of early exposure of isoflurane on chronic pain via the mammalian target of rapamycin signal pathway” by Li et al is interesting and address important issue with Isoflurane use. Manuscript is written well, following are some typo mistakes,
Figure 2, figure labels from (B) to (E) are labeled in capital letters (B-E) while used small letters in figure legends (b-e).
Figure 3 is labeled as Figure 1 in figure legends, and figure labels from (B) to (E) are labeled in capital letters (B-E) while used small letters in figure legends (b-e). Also, written in “italics”.
Figure 4 is labeled as Figure 2 in figure legends, and figure labels (A) and (B) are labeled in capital letters (B-E) while used small letters in figure legends (b-e). Also, written in “italics”.
Author Response
Point 1- Figure 2, figure labels from (B) to (E) are labeled in capital letters (B-E) while used small letters in figure legends (b-e).
Done as suggested.
Point 2- Figure 3 is labeled as Figure 1 in figure legends, and figure labels from (B) to (E) are labeled in capital letters (B-E) while used small letters in figure legends (b-e). Also, written in “italics”.
Done as suggested.
Point 3- Figure 4 is labeled as Figure 2 in figure legends, and figure labels (A) and (B) are labeled in capital letters (B-E) while used small letters in figure legends (b-e). Also, written in “italics”.
Done as suggested.